# Non-Monotonic Latent Alignments for CTC-Based Non-Autoregressive Machine Translation

**Chenze Shao**[1,2], **Yang Feng**[1,2*]
[1]Key Laboratory of Intelligent Information Processing
Institute of Computing Technology, Chinese Academy of Sciences
[2]University of Chinese Academy of Sciences
shaochenze18z@ict.ac.cn, fengyang@ict.ac.cn

## Abstract

Non-autoregressive translation (NAT) models are typically trained with the cross-entropy loss, which forces the model outputs to be aligned verbatim with the target sentence and will highly penalize small shifts in word positions. Latent alignment models relax the explicit alignment by marginalizing out all monotonic latent alignments with the CTC loss. However, they cannot handle non-monotonic alignments, which is non-negligible as there is typically global word reordering in machine translation. In this work, we explore non-monotonic latent alignments for NAT. We extend the alignment space to non-monotonic alignments to allow for the global word reordering and further consider all alignments that overlap with the target sentence. We non-monotonically match the alignments to the target sentence and train the latent alignment model to maximize the F1 score of non-monotonic matching. Extensive experiments on major WMT benchmarks show that our method substantially improves the translation performance of CTC-based models. Our best model achieves 30.06 BLEU on WMT14 En-De with only one-iteration decoding, closing the gap between non-autoregressive and autoregressive models.[2]

## 1 Introduction

Non-autoregressive translation (NAT) models achieve significant decoding speedup in neural machine translation [NMT, 1, 47] by generating target words simultaneously [17]. This advantage usually comes at the cost of translation quality due to the mismatch of training objectives. NAT models are typically trained with the cross-entropy loss, which forces the model outputs to be aligned verbatim with the target sentence and will highly penalize small shifts in word positions. The explicit alignment required by the cross-entropy loss cannot be guaranteed due to the multi-modality problem that there exist many possible translations for the same sentence [17], making the cross-entropy loss intrinsically mismatched with NAT.

As the cross-entropy loss can not evaluate NAT outputs properly, many efforts have been devoted to designing better training objectives for NAT [29, 40–42, 38, 13, 36, 9, 43]. Among them, latent alignment models [29, 36] relax the alignment restriction by marginalizing out all monotonic latent alignments with the connectionist temporal classification loss [CTC, 15], which receive much attention for the ability to generate variable length translation.

Latent alignment models generally make a strong monotonic assumption on the mapping between model output and the target sentence. As illustrated in Figure 1a, the monotonic assumption holds in

---

[*]Corresponding author: Yang Feng
[2]Source code: https://github.com/ictnlp/NMLA-NAT.

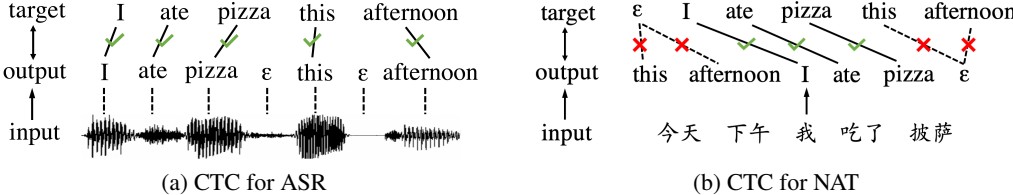

(a) CTC for ASR  (b) CTC for NAT

Figure 1: Illustration of the monotonic alignment assumption of CTC: (a) CTC for ASR where there is a natural monotonic mapping between the speech input and ASR target, (b) CTC for NAT where there exists global word reordering that induces non-monotonic alignment. $\epsilon$ is the blank token that means 'output nothing'.

classic application scenarios of CTC like automatic speech recognition (ASR) as there is a natural monotonic mapping between the speech input and ASR target. However, non-monotonic alignments are non-negligible in machine translation as there is typically global word reordering, which is a common source of the multi-modality problem. As Figure 1b shows, when the target sentence is "I ate pizza this afternoon" but the model produces a translation with a different but correct word ordering "this afternoon I ate pizza", the CTC loss cannot handle this non-monotonic alignment between output and target and wrongly penalizes the model.

In this paper, we propose to model non-monotonic latent alignments for non-autoregressive machine translation. We first extend the alignment space from monotonic alignments to non-monotonic alignments to allow for the global word reordering in machine translation. Without the monotonic structure, we have to optimize the best alignment found by the Hungarian algorithm [46, 28, 7, 9] since it becomes difficult to marginalize out all non-monotonic alignments with dynamic programming. This difficulty can be overcome by not requiring an exact match between alignments and the target sentence. In practice, it is not necessary to have the translation include exact words as contained in the target sentence, but it would be favorable to have a large overlap between them. Therefore, we further extend the alignment space by considering all alignments that overlap with the target sentence. Specifically, we are interested in the overlap of n-grams, which is the core of some evaluation metrics (e.g., BLEU). We accumulate n-grams from all alignments regardless of their positions and non-monotonically match them to target n-grams. The latent alignment model is trained to maximize the F1 score of n-gram matching, which reflects the translation quality to a certain extent [32].

We conduct experiments on major WMT benchmarks for NAT (WMT14 En↔De, WMT16 En↔Ro), which shows that our method substantially improves the translation performance and achieves comparable performance to autoregressive Transformer with only one-iteration parallel decoding.

## 2 Background

### 2.1 Non-Autoregressive Machine Translation

[17] proposes non-autoregressive neural machine translation, which achieves significant decoding speedup by generating target words simultaneously. NAT breaks the dependency among target tokens and factorizes the joint probability of target words in the following form:

$$p(y|x, \theta) = \prod_{t=1}^{T} p_t(y_t|x, \theta), \tag{1}$$

where $x$ is the source sentence belonging to the input space $\mathcal{X}$, $y = \{y_1, ..., y_T\}$ is the target sentence belonging to the output space $\mathcal{Y}$, and $p_t(y_t|x, \theta)$ indicates the translation probability in position $t$.

The vanilla-NAT has a length predictor that takes encoder states as input to predict the target length. During the training, the target length is set to the golden length, and the vanilla-NAT is trained with the cross-entropy loss, which explicitly aligns model outputs to target words:

$$\mathcal{L}_{CE}(\theta) = -\sum_{t=1}^{T} \log(p_t(y_t|x, \theta)). \tag{2}$$

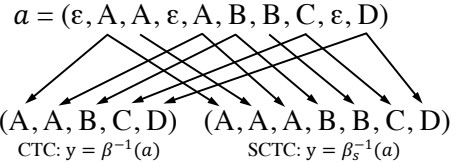

$a = (\varepsilon, A, A, \varepsilon, A, B, B, C, \varepsilon, D)$

$(A, A, B, C, D) \quad (A, A, A, B, B, C, D)$

CTC: $y = \beta^{-1}(a)$ $\qquad$ SCTC: $y = \beta_s^{-1}(a)$

Figure 2: An example of target sentences obtained from collapsing functions of CTC and SCTC. The collapsing function of SCTC is $\beta_s^{-1}$, which only removes blanks in the alignment $a$, where the collapsing function of CTC removes repetitions first and then removes blanks.

## 2.2 NAT with Latent Alignments

The vanilla-NAT suffers from two major limitations. The first limitation is the explicit alignment required by the cross-entropy loss, which cannot be guaranteed due to the multi-modality problem and therefore leads to the inaccuracy of the loss. The second limitation is the requirement of target length prediction. The predicted length may not be optimal and cannot be changed dynamically, so it is often required to use multiple length candidates and re-score them to produce the final translation.

The two limitations can be addressed by using CTC-based latent alignment models [15, 29, 36], which extend the output space $\mathcal{Y}$ with a blank token $\epsilon$ that means 'output nothing'. We define the extended output space as $\mathcal{Y}^*$. Following prior works [14, 36], we refer to the elements $a \in \mathcal{Y}^*$ as alignments, since the location of the blank tokens determines an alignment between the extended output and target sentence. Assume the generated alignments have length $T_a$, we define a function $\beta(y)$ which returns a subset of $\mathcal{Y}^*$ including all possible alignments of length $T_a$ for $y$, where $\beta^{-1} : \mathcal{Y}^* \mapsto \mathcal{Y}$ is the collapsing function that first collapses all consecutive repeated words in $a$ and then removes all blank tokens to obtain the target sentence. As illustrated in Figure 2, the alignment $a = (\epsilon, A, A, \epsilon, A, B, B, C, \epsilon, D)$ is collapsed to the target sentence $y = (A, A, B, C, D)$ with a monotonic mapping from target positions to alignment positions. During the training, latent alignment models marginalize all alignments with the CTC loss [15]:

$$\log p(y|x, \theta) = \log \sum_{a \in \beta(y)} p(a|x, \theta), \tag{3}$$

where the alignment probability $p(a|x, \theta)$ is modeled by a non-autoregressive Transformer:

$$p(a|x, \theta) = \prod_{t=1}^{T_a} p_t(a_t|x, \theta). \tag{4}$$

NAT models with latent alignments achieve superior performance by overcoming the two major limitations of NAT [29, 36, 16, 25, 33, 49]. The weakness in handling non-monotonic latent alignments is noticed in previous work [36, 16] but remains unsolved.

## 3 Approach

In this section, we explore non-monotonic latent alignments for NAT. We introduce a simplified CTC loss in section 3.1, which has a simpler structure that helps further analysis and derivation under the regular CTC loss. We explore non-monotonic alignments under the simplified CTC loss in section 3.2, and utilize the results to derive non-monotonic alignments under the regular CTC loss in section 3.3. Finally, we introduce the training strategy to combine monotonic and non-monotonic latent alignments in section 3.4.

### 3.1 Simplified Connectionist Temporal Classification

In CTC, the collapsing function $\beta^{-1}$ maps an alignment to a target sentence in two steps: (1) collapses all consecutive repeated words, (2) removes all blank tokens. The two operations make the alignment space $\beta(y)$ complex. Therefore, we first introduce the simplified connectionist temporal classification loss (SCTC), which has a simpler structure that helps the derivation of non-monotonic alignments, and the results on SCTC are helpful for further analysis under the regular CTC loss.

To simplify the alignment structure, we consider only using one operation in the collapsing function. One option is only collapsing all consecutive repeated words in the alignment. The concern is that it will limit the expressive power of the model. For example, the target sentence $y = (A, A, B, C, D)$ has probability 0 since it has repeated words that cannot be mapped from any alignment. Therefore, we favor another option that only removes all blank tokens. In this way, the expressive power stays the same and the alignment space becomes much simpler, where alignments of $y$ simply contain $T$ target words and $T_a - T$ blank tokens.

We denote this simplified loss as SCTC, and illustrate the difference between CTC and SCTC in Figure 2. The alignment space for the target sentence $y$ is defined as $\beta_s(y)$, and the collapsing function is defined as $\beta_s^{-1}$. We can still obtain the translation probability $p(y|x, \theta)$ with dynamic programming. We define the forward variable $\alpha_t(s)$ to be the total probability of $y_{1:s}$ considering all possible alignments $a_{1:t}$, which can be calculated recursively from $\alpha_{t-1}(s)$ and $\alpha_{t-1}(s-1)$:

$$\alpha_t(s) = \alpha_{t-1}(s-1)p_t(y_s|x, \theta) + \alpha_{t-1}(s)p_t(\epsilon|x, \theta). \tag{5}$$

Finally, the total translation probability $p(y|x, \theta)$ is given by the forward variable $\alpha_{T_a}(T)$, so we can train the model with the cross-entropy loss $\mathcal{L}_{sctc}(\theta) = -\log \alpha_{T_a}(T)$.

## 3.2 Non-Monotonic Alignments under SCTC

### 3.2.1 Bipartite Matching

In this section, we explore non-monotonic alignments under the SCTC loss, which are helpful for further analysis under the regular CTC loss. We first extend the alignment space from monotonic alignments $\beta_s(y)$ to non-monotonic alignments $\gamma_s(y)$ to allow for the global word reordering in machine translation, where $\gamma_s(y)$ is defined as:

$$\gamma_s(y) = \bigcup_{y' \in P(y)} \beta_s(y'). \tag{6}$$

In the above definition, $P(y)$ represents all permutations of $y$. For example, $P((A, B)) = \{(A, B), (B, A)\}$. By enumerating $P(y)$, we consider all possible word reorderings of the target sentence. Ideally, we want to traverse all alignments in $\gamma_s(y)$ to calculate the log-likelihood loss:

$$\mathcal{L}_{sum}(\theta) = -\log \sum_{a \in \gamma_s(y)} p(a|x, \theta). \tag{7}$$

However, without the monotonic structure, it becomes difficult to marginalize out all latent alignments with dynamic programming. Alternatively, we can minimize the loss of the best alignment, which is an upper bound of Equation 7:

$$\mathcal{L}_{max}(\theta) = -\log \max_{a \in \gamma_s(y)} p(a|x, \theta). \tag{8}$$

Following prior work [46, 7, 9], we formulize finding the best alignment as a maximum bipartite matching problem and solve it with the Hungarian algorithm [28]. Specifically, we observe that the alignment space $\gamma_s(y)$ is simply permutations of $T$ target words and $T_a - T$ blank tokens. Therefore, finding the best alignment is equivalent to finding the best bipartite matching between the $T_a$ model predictions and the $T$ target words plus $T_a - T$ blank tokens, where the two sets of nodes are connected by edges with the prediction log-probability as weights.

### 3.2.2 N-Gram Matching

Without the monotonic structure desired for dynamic programming, calculating Equation 7 becomes difficult. This difficulty is also caused by the strict requirement of exact match for alignments, which makes it intractable to simplify the summation of probabilities. In practice, it is not necessary to force the exact match between alignments and the target sentence since a large overlap is also favorable. Therefore, we further extend the alignment space by considering all alignments that overlap with the target sentence. Specifically, we are interested in the overlap of n-grams, which is the core of some evaluation metrics (e.g., BLEU).

We propose a non-monotonic and non-exclusive n-gram matching objective based on SCTC to encourage the overlap of n-grams. Following the underlying idea of probabilistic matching [39], we introduce the probabilistic variant of the n-gram count to make the objective differentiable.

Specifically, we use $C_g(y)$ to denote the occurrence count of n-gram $g = (g_1, ..., g_n)$ in the target sentence $y$, and use $C_g^s(\theta)$ to denote the probabilistic count of n-gram $g$ for the model with input $x$ and parameter $\theta$, where the superscript $s$ indicates SCTC. For simplicity, we omit the source sentence $x$ in $C_g^s(\theta)$ and the following notations. The probabilistic n-gram count is obtained by accumulating n-grams from all possible target sentences:

$$C_g^s(\theta) = \sum_{y \in \mathcal{Y}} p(y|x, \theta) C_g(y) = \sum_{a \in \mathcal{Y}^*} p(a|x, \theta) C_g(\beta_s^{-1}(a)). \tag{9}$$

The calculation of $C_g^s(\theta)$ is the core of this method and will be described in detail later. We use $M_g^s(\theta)$ to denote the match count of n-gram g between the latent alignment model and the target sentence $y$ (we omit $y$ in $M_g^s(\theta)$ for simplicity), which is defined as follows:

$$M_g^s(\theta) = \min(C_g(y), C_g^s(\theta)). \tag{10}$$

Our objective is to maximize the n-gram overlap between the target sentence and model output, so we can train the model to maximize the precision or recall of n-gram matching:

$$P_n^s(\theta) = \frac{\sum_{g \in G_n} M_g^s(\theta)}{\sum_{g \in G_n} C_g^s(\theta)}, R_n^s(\theta) = \frac{\sum_{g \in G_n} M_g^s(\theta)}{\sum_{g \in G_n} C_g(y)}, \tag{11}$$

where we use $G_n$ to denote the set of all n-grams. However, since latent alignment models have the ability to control the translation length, both precision and recall cannot accurately reflect the translation quality. The precision will encourage short translations to reduce the denominator, while the recall prefers long translations. Therefore, we consider both of them and maximize the F1 score:

$$\mathcal{L}_n^s(\theta) = -\text{F1}_n^s(\theta) = -\frac{2 \cdot \sum_{g \in G_n} M_g^s(\theta)}{\sum_{g \in G_n} (C_g(y) + C_g^s(\theta))}, \tag{12}$$

where we use $G_n$ to denote the set of all n-grams. In the numerator of Equation 12, $M_g^s(\theta)$ is non-zero only if $C_g(y)$ is non-zero, so we only need to calculate probabilistic n-gram counts $C_g^s(\theta)$ for n-grams $g$ in the target sentence. In the denominator, $\sum_{g \in G_n} C_g(y)$ equals to the constant $T - n + 1$. Therefore, there are only two tasks left: calculating $C_g^s(\theta)$ and the summation $\sum_{g \in G_n} C_g^s(\theta)$.

For the first task of calculating $C_g^s(\theta)$, we first consider the simple case $n = 1$. The main difference between 1-gram matching and bipartite matching is that 1-gram matching is non-exclusive, which allows us to marginalize out all alignments using the property of non-autoregressive generation. Due to the space limit, we refer readers to Appendix.A for the derivation and directly give the probabilistic 1-gram count as follows:

$$C_g^s(\theta) = \sum_{t=1}^{T_a} p_t(g_1|x, \theta), g \in G_1, \tag{13}$$

where $G_1$ denotes the set of all 1-grams $g = (g_1)$. Both 1-gram matching and bipartite matching are completely non-monotonic, which aim at generating correct target words regardless of the word order. When the target sentence is "I ate pizza this afternoon", they will think that "pizza ate I this afternoon" is also a good translation. We consider 2-gram matching to overcome this limitation, which is also non-monotonic but takes target dependency into consideration. We first introduce a transition matrix $A$ of size $T_a \times T_a$, where $A_{i,j}$ is the probability that all positions between $i$ and $j$ are blank tokens:

$$A_{i,j} = \begin{cases} 0 & j < i + 1 \\ 1 & j = i + 1 \\ \prod_{t=i+1}^{j-1} p_t(\epsilon|x, \theta) & j > i + 1 \end{cases}. \tag{14}$$

Then we give the probabilistic 2-gram count for $g = (g_1, g_2)$ (see Appendix.A for derivation):

$$C_g^s(\theta) = \sum_{i=1}^{T_a} \sum_{j=1}^{T_a} p_i(g_1|x, \theta) A_{i,j} p_j(g_2|x, \theta), g \in G_2. \tag{15}$$

For a larger $n$, the probabilistic n-gram count is:

$$C_g^s(\theta) = \sum_{i_1=1}^{T_a} \sum_{i_2=1}^{T_a} \cdots \sum_{i_n=1}^{T_a} p_{i_1}(g_1|x, \theta) \times \prod_{k=2}^{n} A_{i_{k-1}, i_k} p_{i_k}(g_k|x, \theta), g \in G_n. \tag{16}$$

Equation 15 can be efficiently calculated in matrix form with $\mathcal{O}(T_a^2)$ time complexity. For Equation 16, we need $\mathcal{O}(nT_a^n)$ operations to calculate it by brute force, which is impractical when $n$ is large. We refer readers to Appendix.B for the $\mathcal{O}(nT_a^2)$ algorithm, which is similar to calculating the n-step transition probability of a Markov chain.

For the second task of efficiently calculating the summation of probabilistic n-gram counts $\sum_{g \in G_n} C_g^s(\theta)$, we also consider the simple case $n = 1$ first, where the summation can be calculated by directly summing all probabilities:

$$\sum_g C_g^s(\theta) = \sum_{t=1}^{T_a} \sum_g p_t(g_1|x,\theta), g \in G_1. \tag{17}$$

For $n > 1$, the search space is too big for the direct summation. Fortunately, the definition of $C_g^s(\theta)$ in Equation 9 allows us to safely extend the result of $n = 1$ to a larger $n$ since n-grams count in a sentence is always $n - 1$ less than 1-grams count.

$$\sum_{g \in G_n} C_g^s(\theta) = \sum_{g \in G_1} C_g^s(\theta) - n + 1. \tag{18}$$

In summary, with the above results (Equation 13-18), we are able to efficiently calculate the loss defined in Equation 12 to maximize the F1 score of non-monotonic n-gram matching, which reflects the translation quality to a certain extent.

## 3.3 Non-Monotonic Alignments under CTC

In this section, we explore non-monotonic latent alignments under the regular CTC loss. Under SCTC, we can use both the bipartite matching and n-gram matching to train the latent alignment model. However, bipartite matching is infeasible under the regular CTC loss. Recall that the search space $\gamma_s(y)$ is simply permutations of target words and $T_a - T$ blank tokens, which allows us to find the best alignment with the Hungarian algorithm. However, as consecutive repeated words will be collapsed in the regular CTC loss, alignments in $\gamma(y)$ may contain different words, so the best alignment cannot be found by the Hungarian algorithm.

Fortunately, for n-gram matching, we can utilize previous results of SCTC to help the analysis and derivation under the regular CTC loss. As the output of the collapsing function $\beta_s^{-1}(a)$ simply contains more repeated words than $\beta^{-1}(a)$, we can first identify how much influence these repeated words have, and then remove this influence from the results of SCTC. We first formally define the probabilistic n-gram count under the regular CTC:

$$C_g(\theta) = \sum_{a \in \mathcal{Y}^*} p(a|x,\theta)C_g(\beta^{-1}(a)). \tag{19}$$

We define $R_g(\theta)$ to be the gap bewteen $C_g(\theta)$ and $C_g^s(\theta)$ when $n = 1$, which represents the number of repeated words $g_1$ removed by the collapsing function. $R_g(\theta)$ can be efficiently calculated as follows (see Appendix.C for derivation):

$$R_g(\theta) = C_g^s(\theta) - C_g(\theta) = \sum_{t=1}^{T_a-1} p_t(g_1|x,\theta)p_{t+1}(g_1|x,\theta), g \in G_1. \tag{20}$$

The above equation enables us to efficiently calculate $C_g(\theta)$ and $\sum_g C_g(\theta)$ when $g \in G_1$. For 2-grams $g = (g_1, g_2)$, we divide it into two cases $g_1 = g_2$ and $g_1 \neq g_2$. The collapsing of consecutive repeated words will only remove 2-grams of the first case, where each removed word corresponds to a removed 2-gram. Taking Figure 2 as an example, the output of SCTC $\beta_s^{-1}(a)$ have two more 2-grams $\{(A, A), (B, B)\}$ of the $g_1 = g_2$ case than the output of CTC $\beta^{-1}(a)$, and they contain exactly the same 2-grams of the $g_1 \neq g_2$ case. Therefore, we can directly reuse previous results of SCTC and the number of repeated words $R_g(\theta)$ to calculate the probabilistic 2-gram count:

$$C_g(\theta) = \begin{cases} C_g^s(\theta) - R_g(\theta), & g_1 = g_2 \\ C_g^s(\theta), & g_1 \neq g_2 \end{cases}, g \in G_2. \tag{21}$$

Unfortunately, for $n > 2$, there is no such a clear relationship between $C_g(\theta)$ and $C_g^s(\theta)$, where the CTC output can contain n-grams (for example, $(A, B, C)$ in Figure 2) that the SCTC output does not

have. If we directly formulize $C_g(\theta)$ like in Equation 16, it will be more complex and the $\mathcal{O}(nT_a^2)$ algorithm is no longer applicable. Therefore, we do not use n-gram matching with $n > 2$ under the regular CTC loss. Finally, we define the match count $M_g(\theta)$ similarly as in Equation 10 and train the CTC-based latent alignment model to maximize the F1 score:

$$\mathcal{L}_n(\theta) = -\frac{2 \cdot \sum_{g \in G_n} M_g(\theta)}{\sum_{g \in G_n}(C_g(y)+C_g(\theta))}. \tag{22}$$

### 3.4 Training

In the above, we propose non-monotonic training objectives for the latent alignment model. They need to be combined with monotonic training objectives, otherwise, they may generate translations like "pizza ate I this afternoon". Therefore, we first pretrain the latent alignment model with monotonic training objectives like SCTC or CTC. The model pretrained by SCTC can be finetuned with the bipartite matching objective (Equation 8) or the n-gram matching objective (Equation 12). The model pretrained by CTC can be finetuned with the n-gram matching objective (Equation 22), and the choices of $n$ are limited to $\{1, 2\}$.

## 4 Experiments

### 4.1 Experimental Setup

**Datasets** We evaluate our methods on the most widely used public benchmarks in previous NAT studies: WMT14 English↔German (En↔De, 4.5M sentence pairs) [5] and WMT16 English↔Romanian (En↔Ro, 0.6M sentence pairs) [6]. For WMT14 En↔De, the validation set is *newstest2013* and the test set is *newstest2014*. For WMT16 En↔Ro, the validation set is *newsdev-2016* and the test set is *newstest-2016*. Following prior works, we use tokenized BLEU [32] to evaluate the translation quality. Considering that BLEU might be biased as our method optimizes the n-gram overlap, we also report METEOR [2], which is not directly based on n-gram overlap. We learn a joint BPE model [37] with 32K operations to process the data and share the vocabulary for source and target languages.

**Knowledge Distillation** Following prior works [17], we apply sequence-level knowledge distillation [26] to reduce the complexity of training data. We use Transformer-base setting for the teacher model and train NAT on the distilled data.

**Implementation Details** We adopt Transformer-base [47] as our autoregressive baseline as well as the teacher model. We uniformly copy encoder outputs to construct decoder inputs. For CTC-based NAT models, the length for decoder inputs is $3\times$ as long as the source length. On WMT14 En↔De, we use a dropout rate of 0.2 to train NAT models and use a dropout rate of 0.1 for finetuning. On WMT16 En↔Ro, the dropout rate is 0.3 for both the pretraining and finetuning. We use the batch size 64K and train NAT models for 300K steps on WMT14 En↔De and 150K steps on WMT16 En↔Ro. During the finetuning, we train NAT models for 6K steps with the batch size 256K. All models are optimized with Adam [27] with $\beta = (0.9, 0.98)$ and $\epsilon = 10^{-8}$. The learning rate warms up to $5 \cdot 10^{-4}$ within 10K steps in the pretraining and warms up to $e \cdot 10^{-4}$ within 500 steps in the finetuning. We use the GeForce RTX 3090 GPU to train models and measure the translation latency. We implement our models based on the open-source framework of `fairseq` [MIT License, 31].

**Decoding** For autoregressive models, we use beam search with beam size 5 for the decoding. For Vanilla-NAT, we use argmax decoding to generate the translation. For CTC-based models, we can also use argmax decoding to generate the alignment and then collapse it to obtain the translation. Besides, following [29, 16], we can use beam search decoding combined with a 4-gram language model [20] to find the translation that maximizes:

$$\log p(y|x, \theta) + \alpha \log p_{LM}(y) + \beta \log(|y|), \tag{23}$$

where $\alpha$ and $\beta$ are hyperparameters for the language model score and length bonus. We use a fixed beam size 20 with grid-search $\alpha$, $\beta$. The beam search does not contain any neural network computations and can be implemented efficiently in C++[3].

---

[3]https://github.com/parlance/ctcdecode

Table 1: Performance comparison between baselines and our methods on WMT14 En-De test set.

| | Model | BLEU | METEOR | Speed |
|---|---|---|---|---|
| Base | Transformer | 27.54 | 54.38 | 1.0× |
| | Vanilla-NAT | 19.32 | 45.79 | 15.5× |
| SCTC | SCTC | 25.26 | 52.06 | |
| | +bipartite | 26.13 | 53.17 | |
| | +1-gram | 26.22 | 53.20 | |
| | +2-gram | **26.79** | 53.54 | 14.7× |
| | +3-gram | 26.66 | 53.48 | |
| | +4-gram | 26.62 | 53.43 | |
| CTC | CTC | 26.34 | 53.15 | |
| | +1-gram | 27.16 | 53.95 | 14.7× |
| | +2-gram | **27.57** | **54.50** | |

Table 2: Performance comparison between our models and existing methods. The speedup is measured on WMT14 En-De test set with batch size 1. AT means autoregressive. **Iter.** denotes the number of iterations at inference time. '–' indicates that the result is not reported. '12-1' means the Transformer with 12 encoder layers and 1 decoder layer [24].

| Models | | Iter. | Speed | WMT14 | | WMT16 | |
|---|---|---|---|---|---|---|---|
| | | | | EN-DE | DE-EN | EN-RO | RO-EN |
| AT | Transformer (teacher) | N | 1.0× | 27.54 | **31.57** | **34.26** | **33.87** |
| | Transformer (12-1) | N | 2.6× | 26.09 | 30.30 | 32.76 | 32.39 |
| | + KD | N | 2.7× | **27.61** | 31.48 | 33.43 | 33.50 |
| Non-CTC NAT | NAT-FT [17] | 1 | 15.6× | 17.69 | 21.47 | 27.29 | 29.06 |
| | NAT-REG [48] | 1 | 27.6× | 20.65 | 24.77 | – | – |
| | AXE [13] | 1 | – | 23.53 | 27.90 | 30.75 | 31.54 |
| | GLAT [33] | 1 | 15.3× | 25.21 | 29.84 | 31.19 | 32.04 |
| | Seq-NAT [42] | 1 | 15.6× | 25.54 | 29.91 | 31.69 | 31.78 |
| | AlignART [45] | 1 | 13.4× | 26.40 | 30.40 | 32.50 | 33.10 |
| | CMLM [12] | 10 | – | 27.03 | 30.53 | 33.08 | 33.31 |
| | LevT [18] | 2.05 | 4.0× | 27.27 | – | – | 33.26 |
| | JM-NAT [19] | 10 | – | 27.69 | **32.24** | 33.52 | 33.72 |
| | RewriteNAT [11] | 2.70 | – | **27.83** | 31.52 | **33.63** | **34.09** |
| | CMLMC [22] | 10 | – | 28.37 | 31.41 | 34.57 | 34.13 |
| CTC-based NAT | CTC [29] | 1 | – | 16.56 | 18.64 | 19.54 | 24.67 |
| | Imputer [36] | 1 | – | 25.80 | 28.40 | 32.30 | 31.70 |
| | GLAT+CTC [33] | 1 | 14.6× | 26.39 | 29.54 | 32.79 | 33.84 |
| | REDER [49] | 1 | 15.5× | 26.70 | 30.68 | 33.10 | 33.23 |
| | + beam&reranking | 1 | 5.5× | 27.36 | 31.10 | 33.60 | 34.03 |
| | DSLP [21] | 1 | 14.8× | 27.02 | 31.61 | 34.17 | **34.60** |
| | Fully-NAT [16] | 1 | 16.8× | 27.20 | 31.39 | 33.71 | 34.16 |
| | Imputer [36] | 8 | – | **28.20** | **31.80** | **34.40** | 34.10 |
| Our Implementation | Bag-of-ngrams [41] | 1 | 15.5× | 25.28 | 29.66 | 31.37 | 31.51 |
| | + rescore 5 candidates | 1 | 9.5× | 25.95 | 30.43 | 32.78 | 32.92 |
| | OAXE [9] | 1 | 15.3× | 24.70 | 29.30 | 31.06 | 31.37 |
| | + rescore 5 candidates | 1 | 14.0 × | 25.78 | 30.29 | 32.45 | 32.63 |
| Our work | CTC w/o finetune | 1 | 14.7× | 26.34 | 29.58 | 33.45 | 33.32 |
| | CTC w/ NMLA | 1 | 14.7× | 27.57 | 31.28 | 33.86 | 33.94 |
| | + beam&lm | 1 | 5.0× | **28.35** | **32.27** | **34.72** | **34.95** |
| | DDRS w/o finetune | 1 | 14.7× | 27.18 | 30.91 | 34.42 | 34.31 |
| | DDRS w/ NMLA | 1 | 14.7× | 28.02 | 31.80 | 34.73 | 34.76 |
| | + beam&lm | 1 | 5.0× | **28.63** | **32.65** | **35.51** | **35.85** |

## 4.2 Main Results

We first compare the performance of different non-monotonic training objectives for latent alignment models. Baseline models include the Transformer, vanilla-NAT, SCTC, and CTC, and we finetune latent alignment models with bipartite matching and n-gram matching objectives. In Table 1, we report BLEU and METEOR scores along with the speedup on WMT14 En-De test set.

From Table 1, we have the following observations: (1) Non-monotonic training objectives can effectively improve the performance of latent alignment models, which improves SCTC by 1.53 BLEU and 1.48 METEOR and improves CTC by 1.23 BLEU and 1.35 METEOR, illustrating the importance of non-monotonic alignments. (2) Despite having the same expressive power, SCTC underperforms CTC in the task of non-autoregressive translation. We speculate that it is because of the difference in the distribution of the alignment space. Most target sentences do not contain repeated words, in which case the alignment space of SCTC $\beta_s^{-1}(y)$ is a subset of $\beta^{-1}(y)$, making the ability of SCTC relatively weak. (3) SCTC achieves the best performance with 2-gram finetuning, and then the performance decreases with the increase of n due to the sparsity of higher rank n-grams. It suggests that we do not need a large n and relieves the concern on CTC that we cannot calculate n-gram matching for $n > 2$. (4) The correlation between the reported BLEU and METEOR scores is very strong, suggesting that BLEU is not biased when evaluating our methods. Therefore, we can safely use BLEU to evaluate the translation quality in the following experiments.

We call the 2-gram matching objective NMLA since it represents our best non-monotonic training method for latent alignment models. Besides CTC, we also apply NMLA finetuning on a stronger baseline DDRS [43][4], which trains the CTC model with multiple references. To extend our method to multiple references, we calculate the n-gram counts for each reference and use their average as the target n-gram count. We compare their performance with existing approaches in Table 2. Notably, CTC with NMLA achieves comparable performance to autoregressive Transformer on all benchmarks, demonstrating the effectiveness of our method. On the strong baseline DDRS, NMLA finetuning also brings remarkable improvements and outperforms autoregressive baselines. With the help of beam search and 4-gram language model, NMLA even outperforms other iterative NAT models and the autoregressive Transformer with only one-iteration parallel decoding. Compared to our implementations of prior works that explore non-monotonic matching for NAT [41, 9], we can see that they underperform the CTC baseline even when rescoring 5 candidates, and their speedup after rescoring also lays behind CTC. It demonstrates the importance of exploring non-monotonic matching on latent alignment models, which is more desirable for its flexibility of generating predictions with variable lengths.

As NMLA finetuning brings remarkable improvements over the strong baseline DDRS, we further conduct experiments to verify whether the NMLA is orthogonal to other methods. In Table 3, we reimplement GLAT+CTC (Fully-NAT) [16, 33] based on their open-source code[5] and finetune it with the NMLA objective. Our implementation replaces the SoftCopy mechanism with UniformCopy, which surprisingly improves 0.7 BLEU and achieves 27.07 BLEU on WMT14 En-De test set. Finetuning with NMLA can further improve its performance to 27.75 BLEU, demonstrating the extensibility of NMLA. In Table 4, we apply NMLA on the big version of DDRS to explore its capability on large models. The results show that the performance of DDRS-big can be greatly boosted by NMLA finetuning. Combined with beam search, it achieves 30.06 BLEU on WMT14 En-De, which narrows the performance gap between non-autoregressive and autoregressive models to a new level.

Table 3: BLEU scores of GLAT+CTC and NMLA finetuning on WMT14 En-De test set.

| Model | BLEU | Speed |
|---|---|---|
| GLAT+CTC w/o finetune | 27.07 | 14.7× |
| GLAT+CTC w/ NMLA | 27.75 | 14.7× |
| + beam&lm | 28.46 | 5.0× |

Table 4: BLEU scores of DDRS-big and NMLA finetuning on WMT14 En-De test set.

| Model | BLEU | Speed |
|---|---|---|
| DDRS-big w/o finetune | 28.24 | 14.1× |
| DDRS-big w/ NMLA | 29.11 | 14.1× |
| + beam&lm | 30.06 | 4.8× |

---

[4]https://github.com/ictnlp/DDRS-NAT
[5]https://github.com/FLC777/GLAT

### 4.3 Training Cost

In this section, we will analyze the training cost and show that our method is cost-effective. The additional training cost of our method only comes from the NMLA finetuning. In Table 5, we compare the training cost of CTC and NMLA on WMT14 En-De. We measure the training speed by Words Per Second (WPS) and find that the training speed of NMLA is slower than CTC, which can be attributed to the large calculation cost of the NMLA objective. However, since we only finetune the model for 6K steps, the overall cost of the NMLA finetuning is much smaller than the CTC pretraining. On the whole, the training cost of CTC with NMLA is less than $1.2\times$ compared to the CTC baseline.

Table 5: The comparison of CTC pretraining and NMLA finetuning. 'WPS' means words per second. 'Step' means the number of training steps. 'Batch' means batch size. 'Time' means the training time measured on 8 GeForce RTX 3090 GPUs.

|                | WPS  | Step | Batch | Time  |
|----------------|------|------|-------|-------|
| CTC pretraining | 141K | 300K | 64K   | 26.4h |
| NMLA finetuning | 60K  | 6K   | 256K  | 5.0h  |

## 5  Related Work

[17] first proposes non-autoregressive machine translation to accelerate the decoding process of NMT. The cross-entropy loss is applied to train the Vanilla-NAT, which requires the explicit alignment between the model output and target sentence. The explicit alignment cannot be guaranteed due to the multi-modality problem, and many efforts have been devoted to mitigating this issue.

One research direction is to enable explicit alignment by leaking part of the target-side information during the training. [17] uses fertilities as a latent variable to specify the number of output words corresponding to each input word. [23, 35, 4] apply vector quantization to train NAT with discrete latent variables, and [30, 44, 16] apply variational inference to train NAT with continuous latent variables. [34, 3, 45] also introduce latent variables that carry the reordering information. Besides using latent variables, [12] directly feeds the partially masked target sentence as the decoder input, and [33] applies a glancing sampling strategy to control the mask ratio.

Another research direction is to propose better training objectives for NAT. [48] introduces two auxiliary regularization terms to reduce errors of repeated translations and incomplete translations. [40] directly optimizes sequence-level objectives with reinforcement learning. [21] trains NAT with deep supervision and feeds additional layer-wise predictions. [43] creates a dataset with multiple references and dynamically selects an appropriate reference for the training. Prior to this work, [41, 13, 9] have explored ways to establish an alignment between the model output and target sentence, which can promote the training of NAT. Their limitation is that they are basically built on the framework of Vanilla-NAT, which requires a target length predictor and cannot dynamically adjust the output sequence length. CTC-based latent alignment models [29, 36, 16] are more desirable due to their superior performance and the flexibility of generating predictions with variable lengths. However, latent alignment models have a weakness in handling non-monotonic alignments, which is noticed in previous work [36, 16] and solved in this work.

## 6  Conclusion

Latent alignment models cannot handle non-monotonic alignments during the training, which is nonnegligible in machine translation. In this paper, we explore non-monotonic latent alignments for NAT and propose two matching objectives named bipartite matching and n-gram matching. Experiments show that our method achieves strong improvements on latent alignment models.

## 7  Acknowledgement

We thank the anonymous reviewers for their insightful comments. This work was supported by National Key R&D Program of China (NO.2018AAA0102502).

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
