# OpenReview forum: " Non-Monotonic Latent Alignments for CTC-Based Non-Autoregressive Machine Translation"
_NeurIPS.cc/2022/Conference — NeurIPS 2022 Accept_

### Official Review · Reviewer_NovV · 2022-07-11

**Rating:** 5
**Confidence:** 4
**Soundness:** 3 good
**Presentation:** 2 fair
**Contribution:** 2 fair

**Summary:**

This paper proposed methods to train non-autoregressive NMT models by relaxing the monotonic alignment restriction in the CTC loss. The paper first proposed a simplified CTC (SCTC) which only takes the blank symbol into consideration and does not allow token repetitions. In order to taking non-monotonic alignments into consideration during training, two solutions, i.e., bipartite matching and n-gram matching, are introduced. The n-gram matching for the regular CTC is further explored.

Experiments are conducted on WMT En—De and En—Ro tasks. The results show the proposed bipartite matching and n-gram matching can significantly improve over the SCTC and CTC baseline, where the combination of n-gram and CTC achieves the best performance. Compared with related works, the proposed systems achieves comparable results.


**Questions:**

See the above weaknesses.

**Limitations:**

NAN

**Strengths And Weaknesses:**

Strengths:

The motivation of this paper is reasonable and clear as the regular CTC loss would penalize predictions which are correct but have different word orders.

Experiments show the proposed solutions for calculating non-monotonic alignment losses are effective.


Weaknesses:

Experimental results are not convincing enough. Compared with the existing Fully-NAT, the proposed NMLA systems achieves comparable results but slower speed. Although the proposed method show improvements over naïve baselines, whether the proposed method could improve Fully-NAT further as an addition is unknown which uses the CTC loss as well during training.

The paper writing should be improved with more clear descriptions of the proposed methods. For example, Line 109 introduces a forward variable which is not mentioned anywhere else. More detailed introduction on CTC may be needed as a comparison so that readers would know the differences between CTC and SCTC formally if any. In addition, I did not get the role of the bipartite matching in this paper which seemingly neither shows better results nor help the derivation of n-gram matching for CTC.

---

> ### Author Response · Authors · 2022-08-02
> **Thank you for your review**
>
> We thank the reviewer for raising the questions and for their suggestions on adding the discussion between CTC and SCTC. We have addressed below the concerns. Please let us know whether your concerns are resolved or if you have any more comments, questions, or suggestions.
>
> > Compared with Fully-NAT, NMLA achieves comparable results but slower speed.
>
> Since there are differences in NMLA and Fully-NAT's experimental settings, the comparison can be assessed by the improvements over the CTC baseline: NMLA improves the CTC baseline by 0.99 BLEU over four datasets on average, while the improvement of Fully-NAT is 0.5 BLEU according to [1]. As far as we know, the reasons why our CTC baseline is slightly weaker than the baseline of Fully-NAT can be attributed to two settings. First, our batch size is 64K, and Fully-NAT uses a batch size of 128K. Their setting can boost the results, but it doubles the training cost. Second, we use the original WMT14 En-De dataset with 4.5M sentence pairs to be consistent with most other methods, whereas Fully-NAT uses a filtered dataset with 4M sentence pairs. The filtered dataset also slightly improves the BLEU scores but is less comparable since most previous works use the original dataset as we do.
>
> Regarding the decoding speed, neither Fully-NAT nor NMLA affects the decoding process, so they theoretically have the same decoding speed. The difference in the reported results is mainly attributed to the different GPU devices, where NMLA uses GeForce RTX 3090 to measure the speedup and Fully-NAT uses Nvidia V100.
>
> [1] Gu et al. Fully non-autoregressive neural machine translation: Tricks of the trade. 2021.
>
> > Whether the proposed method could improve Fully-NAT further as an addition is unknown.
>
> Thank you for raising this point! We have just finished the experiment of Fully-NAT+NMLA (Appendix H, Table 8). Experimental results show that NMLA is still very effective on the strong baseline, which achieves 27.75 BLEU on WMT14 En-De after finetuning Fully-NAT.
>
> > Line 109 introduces a forward variable which is not mentioned anywhere else.
>
> Thank you for pointing it out! We strengthen the use of the forward variable in the revised paper (lines 111-112) and hope it becomes clear now. The forward variable helps us to calculate the probability of the target sentence, so we can calculate the cross-entropy loss to pretrain the model. It is not mentioned elsewhere since we do not need the forward variable in bipartite matching and n-gram matching.
>
> > More introduction to show the differences between CTC and SCTC
>
> Thank you for the suggestion! The only difference between CTC and SCTC lies in the collapsing function. The collapsing function of CTC first collapses all consecutive repeated words first and then removes all blank tokens, whereas the collapsing function of SCTC is simplified and only removes blank tokens. We have updated the paper to emphasize this difference in the caption of Figure 2.
>
> > bipartite matching
>
> We agree that the bipartite matching neither shows better results nor helps the derivation of n-gram matching for CTC. We discuss bipartite matching since it is an important attempt at invoking non-monotonic alignment, which is an intuitive approach and has a good non-monotonic structure. Besides, though bipartite matching underperforms n-gram matching, the improvement it brings is also considerable, which demonstrates the importance of non-monotonic alignment.

---

> > ### Comment · Reviewer_NovV · 2022-08-08
> > **Solid experiments are important**
> >
> > Thank the authors for the response and additional experiments. It is good to see that the proposed method could further improve strong baselines. Accordingly, I raised my score to 5.
> >
> > But I still think the Table 2 is problematic. In addition to the questions on Fully-NAT, another confusion is that GLAT+CTC is only comparable to your CTC baseline, especially given that it is reimplemented by yourself. This means GLAT has a negatively impact on the CTC baseline? Similar to the OAXE. I suggest the authors to improve the comparison and give necessary explanations on experimental results. Rather than simply presenting a single final score, it is import to show the advantages of your methods over other related works based on the same or comparable baselines.
> >
> > PS: I do not think “the comparison can be assessed by the improvements over the CTC baseline” is meaningful.

---

> > > ### Author Response · Authors · 2022-08-08
> > > **Thank you for your suggestions**
> > >
> > > We thank the reviewer for the constructive comments and suggestions! We agree that Table 2 causes some misunderstandings since we did not report the corresponding baseline performance of our re-implementations. We will fix this problem and give clear explanations  in the next version. The misunderstandings are clarified below.
> > >
> > > > GLAT+CTC
> > >
> > > We re-implement GLAT+CTC based on their open-source code and obtain the similar performance with the official results, but it is not directly comparable with our CTC baseline since some settings (e.g., dropout) are different. Thank you for pointing out this problem! It will be fixed by unifying their settings.
> > >
> > > > OAXE
> > >
> > > The baseline model of OAXE is one-iteration CMLM [1], whose performance is much weaker than CTC, so it is reasonable that OAXE underperforms the CTC model. We will report its baseline performance in the next version and show that OAXE achieves significant improvements over its own baseline.
> > >
> > > [1] Ghazvininejad et al. Mask-Predict: Parallel Decoding of Conditional Masked Language Models. 2019.

---

### Official Review · Reviewer_u9nE · 2022-07-11

**Rating:** 7
**Confidence:** 2
**Soundness:** 3 good
**Presentation:** 3 good
**Contribution:** 3 good

**Summary:**

The multi-modality problem in Non-Autoregressive Machine Translation (NAT) makes non-monotonic alignment an important research direction. This paper explores the non-monotonic matching on latent alignment models. The authors first introduce a Simpliﬁed Connectionist Temporal Classiﬁcation (SCTC) where only blank tokens are removed in the collapsing function. Then they derive bipartite matching and n-gram matching as two non-monotonic alignments under SCTC. Finally, they generalize the n-gram matching method under SCTC to regular CTC. The experimental results show the superiority of non-monotonic training objectives, and the best method achieves comparable performance to AT models.

**Questions:**

What best method does NMLA stand for specifically? CTC+2-gram?

**Limitations:**

No majot limitations.

**Strengths And Weaknesses:**

**Strengths**

* The motivation is clearly given and the paper is well-written.
* The proposed non-monotonic alignment methods may facilitate the development of this field.

**Weaknesses**

The relationship of this paper to related work is not very clear. What specific problems exist in the previous works are not sufficiently described.

Typo (line 108): the s translation ->  the translation ?

---

> ### Author Response · Authors · 2022-08-02
> **Thank you for your review**
>
> We thank the reviewer for the positive comments and constructive suggestions! We address each point below and have revised the paper accordingly. Please let us know whether your concerns are resolved or if you have any more comments, questions, or suggestions.
>
> > The relationship of this paper to related work. What specific problems exist in the previous works are not sufficiently described.
>
> We have revised the paper (lines 299-306) to clarify the relationship with related work. Prior to this work, [1,2,3] have explored ways to establish an alignment between the model output and target sentence. Their limitation is that they are basically built on the framework of Vanilla-NAT, which requires a target length predictor and cannot dynamically adjust the output sequence length. CTC-based latent alignment models [4,5,6] are more desirable due to their superior performance and the flexibility of generating predictions with variable lengths. However, latent alignment models have a weakness in handling non-monotonic alignments, which is noticed in previous works [5,6] and solved in this work.
>
> [1] Ghazvininejad et al. Aligned cross entropy for non-autoregressive machine translation. 2020.
>
> [2] Du et al. Order-agnostic cross entropy for non-autoregressive machine translation. 2021.
>
> [3] Shao et al. Minimizing the bag-of-ngrams difference for non-autoregressive neural machine translation. 2020.
>
> [4] Libovický et al. End-to-end non-autoregressive neural machine translation with connectionist temporal classification. 2018.
>
> [5] Saharia et al. Non-autoregressive machine translation with latent alignments. 2020.
>
> [6] Gu et al. Fully non-autoregressive neural machine translation: Tricks of the trade. 2021.
> > Typo
>
> Thank you for correcting the typo! We have updated the paper to fix it.
>
> > What best method does NMLA stand for specifically? CTC+2-gram?
>
> Thank you for the confirmation! You are correct that NMLA stands for the CTC+2-gram method. We have updated the paper (line 275) to make it explicit to the reader.

---

> > ### Comment · Reviewer_u9nE · 2022-08-08
> > **Thank you for your response**
> >
> > The authors' response solved my question.

---

### Official Review · Reviewer_bcDR · 2022-07-11

**Rating:** 7
**Confidence:** 4
**Soundness:** 3 good
**Presentation:** 3 good
**Contribution:** 3 good

**Summary:**

This paper proposes relaxing CTC-based models, which train the model with the objectives that allow non-monotonic alignment between the output and the reference.  Furthermore, the authors propose to finetune a CTC with n-gram matching (n=1, 2). Experiments show that such approaches consistently improve performance, showing the effectiveness of relaxing objectives and finetuning strategy.

**Questions:**

n/a

**Ethics Review Area:**

["I don’t know"]

**Strengths And Weaknesses:**

Overall, I think the paper is publishable, which has:
- The motivation for allowing non-monotonic alignment of the CTC model is clear.
- There is a systematic study on invoking non-monotonic alignment, and the authors show consistent improvements over receptive baselines.
- Their best variant achieves strong performance.
- The proposed SCTC provides a bridge to n-gram matching for the general CTC model, which is novel.

---

> ### Author Response · Authors · 2022-08-02
> **Thank you for your review**
>
> We sincerely appreciate the reviewer’s positive comments! Please let us know if there are any further questions/concerns, and we will be happy to address them.

---

### Meta-Review · Area_Chair_CACQ · 2022-08-21

**Recommendation:** Accept
**Confidence:** Certain

**Metareview:**

This paper proposes relaxing CTC-based models, which train the model with the objectives that allow non-monotonic alignment between the output and the reference. Although there is some concern about the empirical results. Most reviewers think the motivation is clearly given and the paper is well-written.

**Award:**

No

---

### Decision · Program_Chairs · 2022-09-14

Accept